# Energy Management Method for Fast-Charging Stations with the Energy Storage System to Alleviate the Voltage Problem of the Observation Node

**Rui Ye \*, Xueliang Huang and Zexin Yang**

School of Electrical Engineering, Southeast University, Nanjing 210096, China; xlhuang@seu.edu.cn (X.H.); m15695570194@163.com (Z.Y.)
* Correspondence: yerui8500@126.com

**Abstract:** Large-scale fast charging of electric vehicles (EVs) probably causes voltage deviation problems in the distribution network. Installing energy storage systems (ESSs) in the fast-charging stations (FCSs) and formulating appropriate active power plans for ESSs is an effective way to reduce the local voltage deviation problem. Some deterministic centralized strategies used for ESSs at FCSs are proposed to solve the voltage deviation problem mentioned above. However, the randomness of the EV load is very large, which can probably reduce the effects of deterministic centralized strategies. A fast and reliable centralized strategy considering the randomness of the EV load for ESSs is a key requirement. Therefore, we propose in this paper a day-ahead scheduling strategy with the aim of maximizing the probability of the nodal voltage change being smaller than a preset limit at the observation node. In the proposed strategy, the uncertainty of EV load is taken into account and the probability of the voltage change of an observation node is quantified by a proposed analytic assessment model (AMM). Furthermore, a voltage change optimization model (VCOM) based on a novel control parameter $\beta$ is proposed, where $\beta$ can be used as a constraint to suppress the nodal voltage change at the observation node. Finally, the IEEE 33-bus test system is used to verify the effectiveness of the proposed day-ahead ESS strategy.

**Keywords:** electrical vehicles; distribution network; local voltage regulation

## 1. Introduction

The adoption of EVs is regarded as having very large potential in terms of reducing carbon emissions [1,2]. The take-up rate of EVs has increased rapidly in recent years, the number of EVs registered worldwide by 2018 having surpassed 5 million [3]. The high penetration level of EVs can bring on a sharp increase in power load, which may cause adverse impacts on the distribution grid [4], especially fast-charging EVs [5]. During peak load periods, the distribution grid may easily suffer security risks such as transformer overload, voltage problems, or cable overload [6].

The voltage problems caused by increasing EV charging load are widely seen as concerning. Previous studies have confirmed that high penetrations of EVs can reduce the voltage stability margin and may lead to the voltage instability of the power grid [7]. A similar conclusion was obtained by using the IEEE 3-bus test system [8]. Since there is very little voltage instability in the distribution networks operating under normal conditions, more research focus is required on the impacts of EV charging on voltage change. The randomness and impulsive characteristics of EV recharging loads mean that the injection of power at the charging node can vary greatly, which may increase the nodal voltage change at the local node or neighboring nodes. The findings in [9] show that voltage violation may occur at 10–20% EV penetrations, and in [10] the voltage deviation reaches 3.46% at 20% EV penetrations. Furthermore, large-scale EV fast charging also can cause voltage flicker in the distribution network [11].

Many studies show that installing energy storage systems (ESS) in fast-charging stations with an appropriate energy management strategy can alleviate voltage problems. A compensation control strategy for the ESS is proposed to deal with energy imbalance in continuous operation [12]. An advanced ESS control strategy based on fuzzy control is used to reduce the local voltage fluctuation [13]. The authors of Ref. [14] studied the ESS used in the FCS, which demonstrates that the voltage profile of the transformer in the FCS can be improved by the ESS. A local control algorithm for the FCS with flywheel energy storage systems was proposed in [15], which can also guarantee the charging demand power of the EV during the period of charging control. However, the randomness of the EV load is not considered in these studies. Furthermore, the actual adjustable capacity of the ESS in an FCS is very limited so that it is generally less effective despite using the above local voltage regulation methods to solve the voltage problem in the distribution network.

To address these inadequacies, a centralized day-ahead scheduling strategy for ESSs in FCSs is proposed in this paper to reduce the adverse impact of FCSs on the voltage changes at observation nodes in the radial distribution network. This paper assumes that a nodal voltage regulation system has been established between the distribution network operator and the FCS operator to reduce the adverse impact caused by FCSs, which includes a regional-level optimization decision-making system (RODS) and station-level energy management systems (SEMSs), as shown in Figure 1. RODS is responsible for formulating the day-ahead active power plan for the ESS. SEMS $i$ is responsible for executing the day-ahead active power plan $i$.

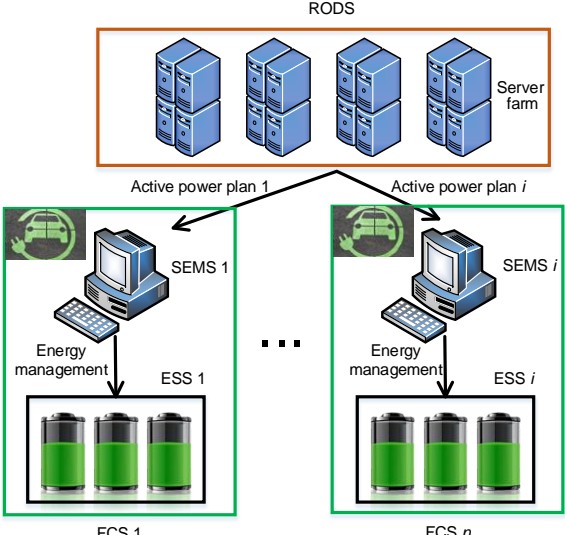

**Figure 1.** The nodal voltage regulation system.

The rest of the paper is arranged as follows. Section 2 presents the problem formulation. Section 3 expounds on the AMM and the control characteristics of the control parameters. The VCOM is illuminated in Section 4. Section 5 presents the case study and discussion. Finally, Section 6 draws conclusions from the research.

## 2. Problem Formulation

### 2.1. FCS Model

The power supply system usually consists of the grid, the photovoltaic system (PV), and the ESS, as shown in Figure 2. Since the traditional load mainly made up of lighting

and air conditioning load is too low in comparison with the EV load, it is reasonable to ignore the traditional load, in which case the FCS model can be expressed as,

$$\begin{cases} P_{j\kappa}^{\mathrm{G}} = P_{j\kappa}^{\mathrm{ch}} - P_{j\kappa}^{\mathrm{S}} \\ P_{j\kappa}^{\mathrm{ch}} = P_{j\kappa}^{\mathrm{EV}} - P_{j\kappa}^{\mathrm{PV}} \\ P_{j\kappa}^{\mathrm{G}} \leq S_{j}^{\mathrm{Trans}} \end{cases} \quad (1)$$

where $P_{j\kappa}^{\mathrm{G}}$ stands for the nodal grid load at the node $j$ at $\kappa$th time interval, $P_{j\kappa}^{\mathrm{PV}}$ is the output power of the photovoltaic (PV) source at $\kappa$th time interval, $P_{j\kappa}^{\mathrm{S}}$ stands for the charge-discharge active power of the ESS at $\kappa$th time interval, where negative sign stands for the discharge and positive sign stands for the charge, $P_{j\kappa}^{\mathrm{EV}}$ is the EV's charging load, and $S_{j}^{\mathrm{Trans}}$ is the rated capacity of the service transformer at the FCS.

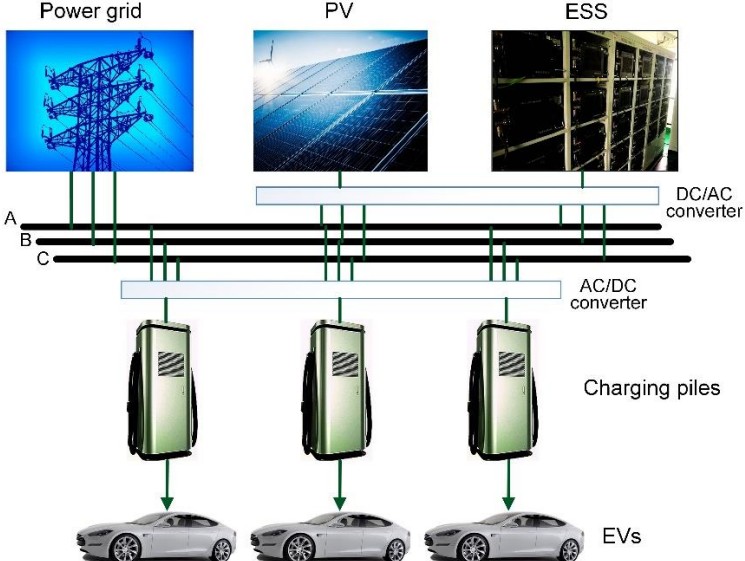

**Figure 2.** Electrical topology diagram of the typical FCS.

In many studies, the nodal load data at different time intervals are regarded as normal distribution variables, and the normal probability density function is usually used to depict the uncertainties of the load of each node [16–18]. We divide a day into 24 equal time intervals, every time interval is 1 h. Let $\Delta P_{j\kappa}^{\mathrm{EV}}$ stand for the EV's charging load change in $\kappa$th time interval, which can be calculated as follows,

$$\Delta P_{j\kappa}^{\mathrm{EV}} = P_{j\kappa}^{\mathrm{EV}} - \mathrm{mean}\left(P_{j\kappa1}^{\mathrm{EV}}, P_{j\kappa2}^{\mathrm{EV}}, \ldots, P_{j\kappa T}^{\mathrm{EV}}\right) \quad (2)$$

Histograms of the EV's charging load change at 4th, 5th, and 7th time intervals can be seen in Figure 3. Although EV charging load changes at these time intervals do not obey normal distribution, they concentrate within 20 kW. Since nodal load changes within 20 kW can hardly affect the nodal voltage significantly under normal circumstances, we use the constant power model to depict the EV's charging load at the time interval when the EV's charging load change is not a normal distribution variable. Let $\kappa_j$ stand for a set of time intervals when the EV's charging load does not obey the normal distribution at the FCS $j$. Then $P_{j\kappa}^{\mathrm{EV}}$ can be formulated as,

$$\begin{cases} P_{j\kappa}^{\mathrm{EV}} \sim N\left(\mu_{j\kappa}, \sigma_{j\kappa}^2\right), \ \kappa \in \kappa_j \\ P_{j\kappa}^{\mathrm{EV}} = \mathrm{mean}(P_{j\kappa1}^{\mathrm{EV}}, P_{j\kappa2}^{\mathrm{EV}}, \ldots, P_{j\kappa T}^{\mathrm{EV}}), \ \kappa \notin \kappa_j \end{cases} \quad (3)$$

where $\mu_{j\kappa}$ and $\sigma_{j\kappa}$ stand for the expectation and standard change of $P_{j\kappa}^{EV}$ respectively. It is reasonable to adopt the constant power model to express the output active power of the PV at the $\kappa$th time interval, since the fact that the change range of the PV power output is usually much smaller than that of the EV's charging load. Hence, when $\kappa \in \kappa_j$, $P_{j\kappa}^{ch}$ is also a normal distribution variable, $N(\mu_{j\kappa} - P_{j\kappa}^{PV}, \sigma_{j\kappa}^2)$. Since the linear transformation of a normal distribution still follows a normal distribution, when $\kappa \in \kappa_j$, the FCS model can be written as,

$$
\begin{cases}
P_{j\kappa}^{G} = \mu_{jk} - P_{j\kappa}^{PV} + \Delta P_{j\kappa}^{ch} - P_{j\kappa}^{S} \\
\Delta P_{j\kappa}^{ch} \sim N\left(0, \sigma_{j\kappa}^2\right) \\
P_{j\kappa}^{G} \le S_j^{Trans} \\
\kappa \in \kappa_j
\end{cases}
\tag{4}
$$

where $\Delta P_{j\kappa}^{ch}$ stands for the Gaussian error component of $P_{j\kappa}^{ch}$.

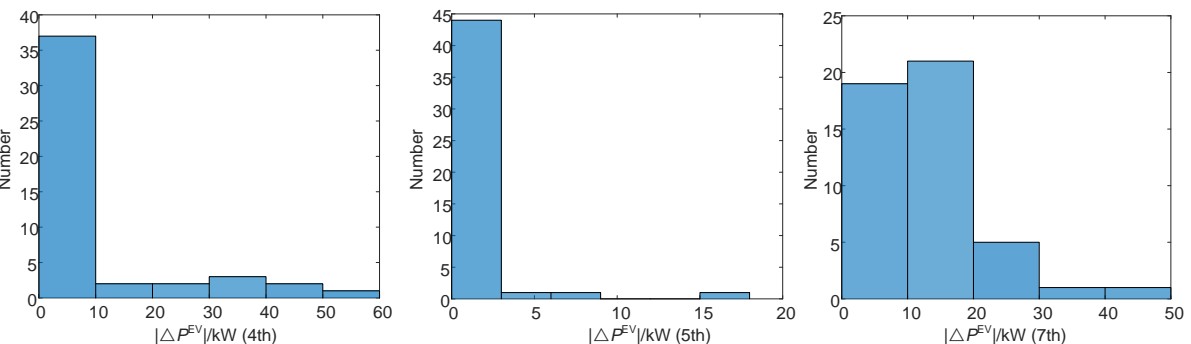

**Figure 3.** Histograms of EV charging load changes at 4th, 5th, and 7th time intervals in a day.

### 2.2. Impact on the Radial Distribution Network

Numerous studies show that large-scale EV fast charging easily leads to nodal voltage change in the radial distribution network, which can deteriorate the power quality. Therefore, it is meaningful to reduce the nodal voltage change caused by FCSs. In this paper, we use the evaluating function $F_{o\kappa}(.)$ to quantify the adverse impact of the FCSs on voltage change at observation node $o$ at $\kappa$th time interval. $F_{o\kappa}(.)$ can be expressed as follows,

$$
\begin{cases}
F_{o\kappa}(\delta_{o\kappa}, \beta_{o\kappa}, \alpha_{o\kappa}) = \frac{1}{\Gamma(\alpha)} \int_0^{\beta_{o\kappa}\delta_{o\kappa}^2} t^{\alpha_{o\kappa}-1} e^{-t} dt \\
\Gamma(\alpha_{o\kappa}) = \int_0^{+\infty} t^{\alpha_{o\kappa}-1} e^{-t} dt, \; \alpha_{o\kappa} > 0
\end{cases}
\tag{5}
$$

where $\delta_{o\kappa}^{o\kappa}$ stands for the upper limit of voltage change at observation node $o$ at $\kappa$th time interval, $\beta_{o\kappa}$ and $\alpha_{o\kappa}$ are control parameter and shape parameter, respectively, and $\Gamma(.)$ is a Gamma function. The detailed descriptions of $F_{o\kappa}(.)$ and the characteristic of the control parameter $\beta_{o\kappa}$ are introduced in Sections 3.2 and 3.3, respectively.

### 2.3. Optimization Model

We can build up a one-objective optimization model to describe the problem which is focused on in this paper,

$$
\begin{aligned}
& \max f(\mathbf{x}) \\
& \text{s.t.} \\
& \quad c_j \le u_j(\mathbf{x}) \le d_j, \; j = 1, \dots, q \\
& \quad a_\kappa \le x_\kappa \le b_\kappa, \; \kappa = 1, \dots, m
\end{aligned}
\tag{6}
$$

where $f(\mathbf{x})$ stands for the objective function calculated by the elements of the optimization vector $\mathbf{x}$, $\mathbf{x} = [x_1, \dots, x_m]$, $u_j(\mathbf{x})$ stands for the $j$th inequality constraint. It is obvious that

many intelligent algorithms can be used to solve Equation (6), such as particle swarm optimization (PSO) and the genetic algorithm (GA). Compared with other intelligent algorithms, the traditional GA is more robust and can effectively search the complex spaces, but it is easier to fall into a local optimum [19–21]. Therefore, an improved real-coded genetic algorithm (RCGA-rdn) is used to solve Equation (6), which can reduce the probability of falling into local optimum and improve the computational efficiency [22]. The constrained optimization model needs to be converted into an unconstrained optimization model before using the RCGA-rdn. In this paper we use the penalty function method to achieve this conversion, then Equation (6) can be rewritten as,

$$
\begin{cases}
\max \left\{ P_1(\mathbf{x}) \cdot f_1(\mathbf{x}), \dots, P_{pp}(\mathbf{x}) \cdot f_{pp}(\mathbf{x}) \right\}, \\
P_g(\mathbf{x}) = 1 - \frac{1}{q} \sqrt{\sum_{j=1}^{q} \left( \frac{\Delta u_{jg}(\mathbf{x})}{\Delta u_j^{\max}(\mathbf{x})} \right)^2} \\
\Delta u_{jg}(\mathbf{x}) = \begin{cases} \left| u_{jg}(\mathbf{x}) - d_j \right| & ,\text{if } u_{jg}(\mathbf{x}) > d_j \\ \left| u_{jg}(\mathbf{x}) - c_j \right| & ,\text{if } u_{jg}(\mathbf{x}) < c_j \\ 0 & ,\text{else} \end{cases} \\
\Delta u_j^{\max}(\mathbf{x}) = \max\left\{ \Delta u_{j1}(\mathbf{x}), \dots, \Delta u_{jpp}(\mathbf{x}) \right\} \\
j = 1, \dots q \\
g = 1, \dots pp
\end{cases}
\tag{7}
$$

where $P_g(.)$ stands for the penalty function of population $g$, and $pp$ is the initial population number. For a detailed description of the optimization model, please refer to Section 4.

### 3. Analytical Assessment Model of the Nodal Voltage Change

*3.1. The Radial Distribution Network Model*

Figure 4 shows a typical radial distribution network, there is only one source node that can be regarded as a voltage source and we use current sources to depict the load nodes. Let $U_s$ stand for the phase voltage at the source node $s$, $U_d$ stand for the phase voltage at the load node $d$, and $U_o$ stand for the phase voltage at the observation node $o$. Based on the circuit superposition theorem, the typical radial distribution network can be decomposed into three different type sub-circuits as shown in Figure 4. The voltage at the observation node $o$ ($U_o$) can be expressed as,

$$
U_o = U_s - \sum_{d \in \mathbf{D}} \frac{S_d^*}{U_d^*} Z_{od}
\tag{8}
$$

where $S_d^*$ stands for the conjugate complex draw power at node $d$, $U_d^*$ is the conjugate phase voltage at node $d$, $Z_{od}$ is the shared impedance between the node $d$ and observation node $o$ from the source node $s$, and $\mathbf{D}$ is a set of the load nodes.

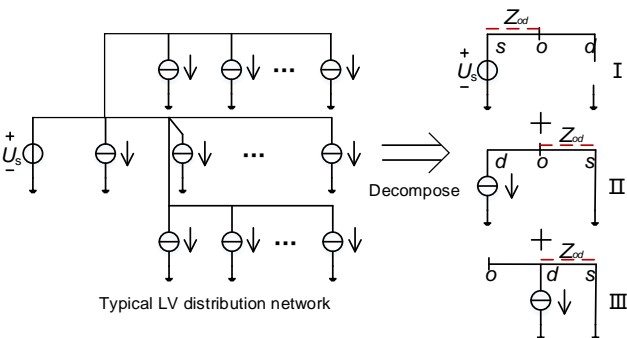

**Figure 4.** The typical LV radial distribution network and circuit decomposition.

Let the complex draw power at the node $d$ change from $S_d$ to $S_d + \Delta S_d$ and the nodal voltage change from $U_d$ to $U_d + \Delta U_d$, then the voltage change at the observation ($\Delta U_o$) can be written as,

$$
\begin{aligned}
\Delta U_o &= -\sum_{d \in \mathbf{D}} \left( \frac{S_d^* + \Delta S_d^*}{U_d^* + \Delta U_d^*} - \frac{S_d^*}{U_d^*} \right) Z_{od} \\
&= \sum_{d \in \mathbf{D}} \frac{S_d^* \Delta U_d^* - \Delta S_d^* U_d^*}{U_d^* (U_d^* + \Delta U_d^*)} Z_{od}
\end{aligned}
\tag{9}
$$

Since $\Delta U_d^* \ll U_d^* (U_d^* + \Delta U_d^*)$, $\Delta U_d^* / U_d^* (U_d^* + \Delta U_d^*) \to 0$, Equation (9) can be simplified as,

$$
\Delta U_o = -\sum_{d \in \mathbf{D}} \frac{\Delta S_d^*}{U_d^* + \Delta U_d^*} Z_{od}
\tag{10}
$$

Let $Z_{od} = R_{od} + jX_{od}$, $\Delta S_d = \Delta P_d^G + j\Delta Q_d^G$ and $\Delta U_d = \Delta U_d^r + j\Delta U_d^i$. By listing the real and imaginary parts of $\Delta U_o$ separately, we can get,

$$
\begin{cases}
\Delta U_o^r = -\sum_{d \in D} \left[ \dfrac{\left(\Delta P_d^G R_{od} + \Delta Q_d^G X_{od}\right)\left(U_d^r + \Delta U_d^r\right)}{\left(U_d^r + \Delta U_d^r\right)^2 + \left(U_d^i + \Delta U_d^i\right)^2} - \dfrac{\left(\Delta P_d^G X_{od} - \Delta Q_d^G R_{od}\right)\left(U_d^i + \Delta U_d^i\right)}{\left(U_d^r + \Delta U_d^r\right)^2 + \left(U_d^i + \Delta U_d^i\right)^2} \right] \\[4mm]
\Delta U_o^i = -\sum_{d \in D} \left[ \dfrac{\left(\Delta P_d^G X_{od} - \Delta Q_d^G R_{od}\right)\left(U_d^r + \Delta U_d^r\right)}{\left(U_d^r + \Delta U_d^r\right)^2 + \left(U_d^i + \Delta U_d^i\right)^2} + \dfrac{\left(\Delta P_d^G R_{od} + \Delta Q_d^G X_{od}\right)\left(U_d^i + \Delta U_d^i\right)}{\left(U_d^r + \Delta U_d^r\right)^2 + \left(U_d^i + \Delta U_d^i\right)^2} \right]
\end{cases}
\tag{11}
$$

where $\Delta U_o^r$ and $\Delta U_o^i$ stand for the real and imaginary parts of $\Delta U_o$. Since the phase angle of the phase voltage is usually small, it is reasonable to ignore the $\Delta U_d^r$ and $\Delta U_d^i$. Let $U_d = |U_d| \angle \theta_d$, then Equation (11) can be rewritten as,

$$
\begin{cases}
\Delta U_o^r \approx \sum\limits_{d \in \mathbf{D}} \left[ A_{od} \Delta P_d^G + B_{od} \Delta Q_d^G \right] \\[2mm]
\Delta U_o^i \approx \sum\limits_{d \in \mathbf{D}} \left[ C_{od} \Delta P_d^G + D_{od} \Delta Q_d^G \right] \\[2mm]
A_{od} = -\dfrac{R_{od} \cos \theta_d - X_{od} \sin \theta_d}{|U_d|}, \; B_{od} = -\dfrac{X_{od} \cos \theta_d + R_{od} \sin \theta_d}{|U_d|} \\[3mm]
C_{od} = -\dfrac{X_{od} \cos \theta_d + R_{od} \sin \theta_d}{|U_d|}, \; D_{od} = \dfrac{R_{od} \cos \theta_d - X_{od} \sin \theta_d}{|U_d|}
\end{cases}
\tag{12}
$$

Since $U_d$ and $Z_{od}$ are known variables, $A$, $B$, $C$, and $D$ are constants. Therefore, $\Delta U_o^r$ and $\Delta U_o^i$ can be approximated as a linear combination of $\Delta P_d^G$ and $\Delta Q_d^G$.

### 3.2. The AMM

It is obvious that the probability of the voltage change ($\Delta U_o$) smaller than the limiting value ($\delta_o$) can be used to measure the impact of the FCSs on the voltage change at the observation node $o$. The larger $P(|\Delta U_o| < \delta_o)$ is, the smaller the impact of the FCSs on the voltage change will be. In this section, we derive the AMM which is used to evaluate the impact of the FCSs on the voltage change at any node in the LV radial distribution network. In this paper, we select the average load at the node without the FCS as the initial load at this node. Let the complex draw power change at the node without the FCS be zero, then we can obtain the nodal voltage change of the observation node, which is caused by charging load changes of FCSs. The reactive power change ($\Delta Q$) can be ignored at the node with the FCS since the power factor of the node with the FCS is usually close to 1 so that the reactive power change is usually very small. Hence, for an $n$-node LV distribution network system, Equation (12) can be rewritten as,

$$
\begin{cases}
\Delta U_o^r = \sum\limits_{i=1}^{n} A_{oi} \Delta P_i^G, \; A_{oi} = -\dfrac{R_{oi} \cos \theta_i - X_{oi} \sin \theta_i}{|U_i|} \\[3mm]
\Delta U_o^i = \sum\limits_{i=1}^{n} C_{oi} \Delta P_i^G, \; C_{oi} = -\dfrac{X_{oi} \cos \theta_i + R_{oi} \sin \theta_i}{|U_i|}
\end{cases}
\tag{13}
$$

where $\Delta P_i^G$ is 0 at the node without the FCS and is $\Delta P_{jk}^G$ at the node with the FCS. Rewrite Equation (13) into a vector form as follows,

$$\begin{bmatrix} \Delta U_o^r \\ \Delta U_o^i \end{bmatrix} = \begin{bmatrix} A_{o1} & \cdots & A_{on} \\ C_{o1} & \cdots & C_{on} \end{bmatrix} \begin{bmatrix} \Delta P_1^G \\ \vdots \\ \Delta P_n^G \end{bmatrix} \tag{14}$$

Let the ESS at the FCS have two output modes as follows:

$$\text{Mode1}: P_{jk}^S = (1 - k_{jk}) P_{jk}^{ch}, \kappa \in \kappa_j \tag{15}$$

$$\text{Mode2}: P_{jk}^S = C_{jk}^S, \kappa \notin \kappa_j \tag{16}$$

where the output active power of the ESS is a normal distribution variable when $\kappa \in \kappa_j$ and is a constant ($C_{jk}^S$) when $\kappa \notin \kappa_j$. By plugging Equation (15) into Equation (1), $P_{jk}^G$ is,

$$P_{jk}^G = k_{jk} P_{jk}^{ch}, \ \kappa \in \kappa_j \tag{17}$$

Since $P_{jk}^{ch}$ is a normal variable when $\kappa \in \kappa_j$, $P_{jk}^G$ is still a normal variable at this time. Let $\mu_{jk}^G$ stand for the expectation of $P_{jk}^G$ when $\kappa \in \kappa_j$. Select $\mu_{jk}^G$ as the initial nodal load for the FCS $j$, then $\Delta P_{jk}^G$ is a normal variable, $\Delta P_{jk}^G \sim N(0, \gamma_{jk}^2)$. By plugging Equation (16) into Equation (1), we can obtain the expression of $P_{jk}^G$ when $\kappa \notin \kappa_j$,

$$P_{jk}^G = P_{jk}^{EV} - P_{jk}^{PV} - C_{jk}^S, \kappa \notin \kappa_j \tag{18}$$

Since $P_{jk}^{PV}$, $P_{jk}^{EV}$, and $C_{jk}$ are all constant when $\kappa \notin \kappa_j$, $\Delta P_{jk}^G$ is zero at this time. Then Equation (14) can be rewritten as,

$$\begin{cases} \begin{bmatrix} \Delta U_{ok}^r \\ \Delta U_{ok}^i \end{bmatrix} = \begin{bmatrix} A_{o1k} & \cdots & A_{onk} \\ C_{o1k} & \cdots & C_{onk} \end{bmatrix} \begin{bmatrix} \Delta P_{1k}^G \\ \vdots \\ \Delta P_{nk}^G \end{bmatrix} \\ \Delta P_{ik}^G \sim N\left(0, \gamma_{jk}^2\right), \kappa \in \kappa_j \&\& i \in \mathbf{E} \\ \Delta P_{ik}^G = 0, \kappa \notin \kappa_j || i \notin \mathbf{E} \end{cases} \tag{19}$$

where $\mathbf{E}$ is a set of nodes with the FCS. According to Equation (19), $\Delta U_{ok}^r$ and $\Delta U_{ok}^i$ are linear combinations of $\Delta P_{jk}^G$. Therefore, $\Delta U_{ok}^r$ and $\Delta U_{ok}^i$ can be regarded as normal distribution variables. We assume that the load at each node is independent of each other, then $\Delta U_{ok}^r$ and $\Delta U_{ok}^i$ can be expressed as,

$$\Delta U_{ok}^r \sim N\left(0, \sum_{j=1}^{n} A_{ojk}^2 \gamma_{jk}^2\right), \text{and } \Delta U_{ok}^i \sim N\left(0, \sum_{j=1}^{n} C_{ojk}^2 \gamma_{jk}^2\right) \tag{20}$$

Since Equation (20) has the same form for each time interval, the subscript $\kappa$ is omitted for convenience of writing in the following derivation. Let $\mathbf{\Delta U}_o = (\Delta U_o^r, \Delta U_o^i)^T$, $\mathbf{\Delta U}_o \sim N(0, \mathbf{C}_o)$, where $\mathbf{C}_o$ is a covariance matrix of $\mathbf{\Delta U}_o$ for observation node $o$. $\mathbf{C}_o$ can be written as,

$$\mathbf{C}_o = \begin{bmatrix} \sum_{j=1}^{n} A_{oj}^2 \gamma_j^2 & \sum_{j=1}^{n} A_{oj} C_{oj} \gamma_j^2 \\ \sum_{j=1}^{n} A_{oj} C_{oj} \gamma_j^2 & \sum_{j=1}^{n} C_{oj}^2 \gamma_j^2 \end{bmatrix} \tag{21}$$

$\mathbf{C}_o$ can be diagonalization by eigenvalue decomposition as:

$$\Lambda_o = \mathbf{W}_o{}^{\mathrm{T}} \mathbf{C}_o \mathbf{W}_o = \begin{bmatrix} \lambda_{1o} & \\ & \lambda_{2o} \end{bmatrix} \tag{22}$$

where $\mathbf{W}_o$ is an eigenmatrix of $\mathbf{C}_o$, $\lambda_{1o}$ and $\lambda_{2o}$ are related eigenvalues and $\mathbf{\Lambda}_o$ is the diagonally similar matrix of $\mathbf{C}_o$. Let $\mathbf{V}_o = W_o^{\mathrm{T}} \Delta \mathbf{U}_o$, where $\mathbf{V}_o = (V_{1o}, V_{2o})^{\mathrm{T}}$. Since $\mathrm{Cov}(V_{1o}, V_{2o})$ is equal to 0, $V_{1o}$ and $V_{2o}$ are independent mutual variables. $V_o^{\mathrm{T}} \mathbf{V}_o$ can be expanded as,

$$\begin{aligned} \mathbf{V}_o^{\mathrm{T}} \mathbf{V}_o \quad &= \Delta \mathbf{U}_o^{\mathrm{T}} \mathbf{W}_o \mathbf{W}_o^{\mathrm{T}} \Delta \mathbf{U}_o \\ &= \left(\Delta U_o^{\mathrm{r}}\right)^2 + \left(\Delta U_o^{\mathrm{i}}\right)^2 \end{aligned} \tag{23}$$

Since $\Delta U_o^2 = (\Delta U_o^{\mathrm{r}})^2 + (\Delta U_o^{\mathrm{r}})^2$, $\Delta U_o^2 = V_o^{\mathrm{T}} \mathbf{V}_o$. Therefore, $\Delta U_o^2$ is the sum of two independent weighed chi-square random variables. Then $\Delta U_o^2$ can approximately obey Gamma distribution ($\Gamma(\alpha_o, \beta_o)$) and the parameters of $\Gamma$ ($\alpha_o, \beta_o$) can be calculated as [22],

$$\alpha_o = \frac{(\lambda_1 + \lambda_2)^2}{2(\lambda_1^2 + \lambda_2^2)}, \text{ and } \beta_o = \frac{\lambda_1 + \lambda_2}{2(\lambda_1^2 + \lambda_2^2)} \tag{24}$$

Then, $\mathrm{P}(\,|\Delta U_o|\, < \delta_o)$ can be calculated as,

$$F_o(\delta_o, \beta_o, \alpha_o) = \mathrm{P}(|\Delta U_o| < \delta_o) = \int_0^{\delta_o^2} \frac{\beta_o^{\alpha_o}}{\Gamma(\alpha_o)} x^{\alpha_o - 1} e^{-\beta_o x} dx \tag{25}$$

The smaller the $F_o(.)$ is, the greater the impact of the FCSs on the voltage change at the observation node $o$ will be.

### 3.3. The Control Characteristic of $\beta_o$

When $\delta_o$ is a known constant, $Fo(.)$ is decided by parameters $\alpha_o$ and $\beta_o$. We can increase $Fo(.)$ by controlling the two parameters. In this section, we discuss the control characteristic of $\beta_o$, which can give theoretical support for the VCOM. According to Equation (24), $\alpha_o$ and $\beta_o$ are functions of $\lambda_{1o}$ and $\lambda_{2o}$. We next note that $\lambda_{1o}$ and $\lambda_{2o}$ can be expanded as,

$$\begin{cases} \lambda_{1o} = \frac{1}{2} \left\{ \sum\limits_{j=1}^{n} K_{oj} \gamma_j^2 + \sqrt{\left\{ \sum\limits_{j=1}^{n} K_{oj} \gamma_j^2 \right\}^2 - 4 \sum\limits_{j=1}^{n-1} Z_{oj} \gamma_{j\kappa}^2 \gamma_{j+1}^2} \right\} \\ \lambda_{2o} = \frac{1}{2} \left\{ \sum\limits_{j=1}^{n} K_{oj} \gamma_j^2 - \sqrt{\left\{ \sum\limits_{j=1}^{n} K_{oj} \gamma_j^2 \right\}^2 - 4 \sum\limits_{j=1}^{n-1} Z_{oj} \gamma_{j\kappa}^2 \gamma_{j+1}^2} \right\} \\ K_{oj} = A_{oj}^2 + C_{oj}^2 \\ Z_{oj} = A_{oj} C_{oj+1} - A_{oj+1} C_{oj} \end{cases} \tag{26}$$

Further expand $Z_{oj}$ and we can obtain the expansion of $Z_{oj}$ as follows,

$$\begin{aligned} Z_{oj} &= A_{oj} C_{oj+1} - A_{oj+1} C_{oj} \\ &= \frac{\left( R_{oj} R_{oj+1} - X_{oj} X_{oj+1} \right) \sin \theta_{jj+1} + \left( R_{oj} X_{oj+1} - R_{oj+1} X_{oj} \right) \cos \theta_{jj+1}}{|V_j||V_{j+1}|} \end{aligned} \tag{27}$$

where $\theta_{jj+1} = \theta_{j+1} - \theta$. Since the phase difference of the voltage between the neighboring nodes in the LV radial distribution network is usually very small, it is reasonable to assume that $\theta_j \approx \theta_{j+1}$, then Equation (22) can be simplified as,

$$Z_{oj} = \frac{R_{oj} X_{oj+1} - R_{oj+1} X_{oj}}{|V_j||V_{j+1}|} \tag{28}$$

The impedance angle difference between shared impedances between adjacent nodes is usually small, therefore,

$$
\begin{aligned}
&\tan \varphi_{oj} - \tan \varphi_{oj+1} = \frac{X_{oj}}{R_{oj}} - \frac{X_{oj+1}}{R_{oj+1}} \to 0 \\
&\Rightarrow Z_j \to 0 \\
&\Rightarrow \left\{ \sum_{j=1}^{n} K_j \gamma_j^2 \right\}^2 \gg 4 \sum_{j=1}^{n-1} Z_j \gamma_{j\kappa}^2 \gamma_{j+1}^2 \\
&\Rightarrow \left\{ \sum_{j=1}^{n} K_j \gamma_j^2 \right\}^2 - 4 \sum_{j=1}^{n-1} Z_j \gamma_{j\kappa}^2 \gamma_{j+1}^2 \to \left\{ \sum_{j=1}^{n} K_j \gamma_j^2 \right\}^2
\end{aligned}
\tag{29}
$$

where $\varphi_{oj}$ is the impedance angle of the shared impedance $Z_{oj}$. Therefore, Equation (26) can be simplified as,

$$
\begin{cases}
\lambda_1 \approx \sum_{j=1}^{n} \left[ A_{oj}^2 + C_{oj}^2 \right] \gamma_j^2 \\
\lambda_2 \approx 0
\end{cases}
\tag{30}
$$

By plugging Equation (30) into Equation (24), we can obtain,

$$
\alpha_o \approx 0.5, \text{ and } \beta_o = \frac{1}{2 \sum_{j=1}^{n} \left[ A_{oj}^2 + C_{oj}^2 \right] \gamma_j^2}
\tag{31}
$$

From Equation (31), $\alpha_o$ is approximately equal to 0.5, which is not affected by the change of the nodal load change. Let $\alpha_o = 0.5$ and $t = \beta_o x$, then $F_o(.)$ can be rewritten as,

$$
F_o(\delta_o, \beta_o) = \int_0^{\beta_o \delta_o^2} \frac{t^{-0.5} e^{-t}}{\Gamma(0.5)} dt
\tag{32}
$$

Since the integrand function of Equation (32) is not less than 0 and $\beta_o > 0$, $F_o(.)$ increases as the increase of $\beta_o$ until $F_o(.)$ reaches 1. In other words, the smaller the value of $\beta_o$ is, the smaller the nodal voltage change at the observation will be.

## 4. The VCOM

We restore the subscript $\kappa$ in the rest of this paper. The objective of the day-ahead scheduling strategy for the ESS is to reduce the adverse impact of the FCSs on nodal voltage change at observation nodes. According to the analysis in Section 3, we can use the AMM to evaluate this impact. Therefore, the optimization objective function can be written as,

$$
F_o(\delta_o, \beta_o) = \int_0^{\beta_o \delta_o^2} \frac{t^{-0.5} e^{-t}}{\Gamma(0.5)} dt
\tag{33}
$$

where **O** is a set of observation nodes. The inequality constraint of $P_{j\kappa}^{S}$ can be written as,

$$
P\left( P_{j\kappa}^{Sd} \leq P_{j\kappa}^{S} \leq P_j^{S+} \right) \geq \xi_j^{Ps}, \kappa \in \kappa_j
\tag{34}
$$

$$
P_j^{S-} \leq P_{j\kappa}^{S} \leq P_j^{S+}, \kappa \in \kappa_j
\tag{35}
$$

where $\xi_j^{Ps}$ is a preset constant for restraining $P_{j\kappa}^{S}$, $P_j^{S+}$, and $P_{j\kappa}^{Sd}$ stand for the rated upper and dynamic lower limits of the charge–discharge active power for the ESS at the FCS $j$, respectively. $P_{j\kappa}^{S-}$ can be expressed as,

$$
P_{j\kappa}^{Sd} =
\begin{cases}
P_{j\kappa}^{ch} - S^{Trans}, & P_{j\kappa}^{ch} - S^{Trans} > P_j^{S-} \\
P_j^{S-}, & P_{j\kappa}^{ch} - S^{Trans} \leq P_j^{S-}
\end{cases}
\tag{36}
$$

where $P_j^{S-}$ is the rated lower limit of the charge-discharge active power for the ESS at the FCS $j$. Let $k_{j\kappa} = \gamma_{j\kappa}/\delta_{j\kappa}$ when $\kappa \in \kappa_j$, we can rewrite Equation (15) by plugging $k_{j\kappa} = \gamma_{j\kappa}/\delta_{j\kappa}$ in Equation (15),

$$P_{j\kappa}^S = \left(1 - \frac{\gamma_{j\kappa}}{\sigma_{j\kappa}}\right) P_{j\kappa}^{ch}, \kappa \in \kappa \tag{37}$$

Next, according to the Bayesian probability formula, the analytical expression for the constraint (34) can be expanded as,

$$\begin{cases} P\left(P_{j\kappa}^{Sd} \le P_{j\kappa}^S \le P_j^{S+}\right) = \begin{cases} \left[\begin{array}{l} \Phi(a_{j\kappa}) - \left(1 - \Phi(b_{j\kappa})\right)\left(1 - \Phi(c_{j\kappa})\right) \\ -\Phi(c_{j\kappa})\Phi(d_{j\kappa}) \ge \xi_j^{Ps} \end{array}\right], \kappa \in \kappa_j \\ \Phi(c_{j\kappa}) + \Phi(e_{j\kappa}) - \Phi(c_{j\kappa})\Phi(e_{j\kappa}) \ge \xi_j^{Ps}, \kappa \notin \kappa_j \end{cases} \\ a_{j\kappa} = \frac{\sigma_{j\kappa} P_j^{S+}}{\sigma_{j\kappa}\left(\sigma_{j\kappa} - \gamma_{j\kappa}\right)} - \frac{\mu_{j\kappa} - P_{j\kappa}^{PV}}{\sigma_{j\kappa}}, b_{j\kappa} = \frac{S_j^{Trans}}{k_\kappa \sigma_{j\kappa}} - \frac{\mu_{j\kappa} - P_{j\kappa}^{PV}}{\sigma_{j\kappa}} \\ c_{j\kappa} = \frac{P_{j\kappa}^{PV} + P_j^{S-} + S^{Trans} - \mu_{j\kappa}}{\sigma_{j\kappa}}, d_{j\kappa} = \frac{\sigma_{j\kappa} P_j^{S-}}{\sigma_{j\kappa}\left(\sigma_{j\kappa} - \gamma_{j\kappa}\right)} - \frac{\mu_{j\kappa} - P_{j\kappa}^{PV}}{\sigma_{j\kappa}} \\ e_{j\kappa} = \frac{P_{j\kappa}^{PV} + P_j^{S-} + S^{Trans} - \mu_{j\kappa}}{\sigma_{j\kappa}} \end{cases} \tag{38}$$

where $\Phi(.)$ is the standard normal probability distribution function. Let $SOC_{j0}$ stand for the initial state of charge (SOC) for the ESS at the FCS $j$ at the beginning of the day, then $SOC_{j\kappa}$ can be written as,

$$SOC_{j\kappa} = SOC_{j0} - \frac{1}{E_j^S}\sum_{i=1}^{\kappa} P_{ji}^S \tag{39}$$

where $E_j^S$ stands for the rated capacity of the ESS at the FCS $j$. The constraint of the SOC can be written as,

$$P\left(SOC_j^- \le SOC_{j\kappa} \le SOC_j^+\right) \ge \xi_j^{SOC} \tag{40}$$

where $\xi_j^{SOC}$ is a preset constant for restraining $SOC_{j\kappa}$. $SOC_j^+$ and $SOC_j^-$ stand for the rated upper and lower limits of the SOC for the ESS at the FCS $j$. The analytical expression of the constraint (40) is,

$$\begin{cases} P\left(SOC_j^- \le SOC_{j\kappa} \le SOC_j^+\right) = \Phi(w_{j\kappa}) - \Phi(r_{j\kappa}) \ge \xi_j^{SOC} \\ w_{j\kappa} = \frac{E_j^S\left(SOC_{j0} - SOC_j^-\right)}{\sigma_Q} - \frac{\sum\limits_{i \notin \kappa_j \&\& i \le \kappa} P_{ji}^S + \mu_Q}{\sigma_Q} \\ r_{j\kappa} = \frac{E_j^S\left(SOC_{j0} - SOC_j^+\right)}{\sigma_Q} - \frac{\sum\limits_{i \notin |\kappa_j| \&\& i \le \kappa} P_{ji}^S + \mu_Q}{\sigma_Q} \\ \mu_Q = \sum\limits_{i=1}^{|\kappa_j \cap T_j^\kappa|} \left(1 - \frac{\gamma_{ji}}{\sigma_{ji}}\right)\left(\mu_{j\kappa} - P_{j\kappa}^{PV}\right) \\ \sigma_Q^2 = \sum\limits_{i=1}^{|\kappa_j \cap T_j^\kappa|} \left(1 - \frac{\gamma_{ji}}{\sigma_{ji}}\right)^2 \sigma_{ji}^2 + 2\left(\sum\limits_{i=1}^{|\kappa_j \cap T_j^\kappa|-1}\sum\limits_{l=i+1}^{|\kappa_j \cap T_j^\kappa|}\left(1 - \frac{\gamma_{ji}}{\sigma_{ji}}\right)\left(1 - \frac{\gamma_{jl}}{\sigma_{jl}}\right)Cov\left(P_{ji}^{ch}, P_{jl}^{ch}\right)\right) \end{cases} \tag{41}$$

where $T_j^\kappa$ is a set of time intervals, $T_j^\kappa = [1, 2, \dots, \kappa]$. Then the basic constraints which are used to ensure the normal operation of the ESS at the FCS $j$ can be listed as follows:

$$\begin{cases} (36), (39) \\ P_j^{S-} \le P_{j\kappa}^S \le P_j^{S+}, \kappa \in \kappa_j \end{cases} \tag{42}$$

Using the analysis according to Section 3.3, we can control the voltage change by

adjusting $\beta_o$, which means $\beta_{o\kappa}$ can be designed as a constraint which is used to restrain the voltage change at the observation node in the optimization framework as follows:

$$\beta_{o\kappa} \geq \beta_{o\kappa}^{\min} \tag{43}$$

where $\beta_{o\kappa}^{\min}$ stands for the lower limit for $\beta_{o\kappa}$. According to Equation (32), $F_{o\kappa}(.)$ increases with the increase of $\beta_{o\kappa}$ until it reaches 1 when $\delta_o$ is a known constant, we can obtain $\beta_{o\kappa}^{\min}$ by the Algorithm 1 as follows:

---

**Algorithm 1: Calculate $\beta_{o\kappa}^{\min}$**

---

1: **INPUT:** This algorithm knows the acceptable probability of the nodal voltage change at the observation node ($F_{o\kappa}^{\min}$) at time interval $\kappa$, the initial nodal load at each node at time interval $\kappa$, the adjustment step size of $\beta_{o\kappa}^{\min}$ ($\Delta$), basic information of the distribution network, which can be used to calculate power flow by using traditional Newton power flow method.
2: **OUTPUT:** $\beta_{o\kappa}^{\min} = [\beta_{1\kappa}^{\min}, \beta_{2\kappa}^{\min}, \dots, \beta_{n\kappa}^{\min}]$
3: **PROCEDURE:**
4: Obtain the nodal voltage at each node by using traditional Newton power flow method.
4: **for** $o = 1$ to $n$ **do**
5: Obtain $\beta_{o\kappa}$ according to Equation (31)
6: Obtain $F_{o\kappa}(.)$ according to Equation (32)
7: **for** $I = 1$ to $N$ **do**
8: **if** $F_{o\kappa}(.) \geq F_{o\kappa}^{\min}$
9: $\beta_{o\kappa}^{\min} \leftarrow \beta_{o\kappa} - \Delta$
10:   Update $F_{o\kappa}(.)$ by plugging $\beta_{o\kappa}^{\min}$ into Equation (32)
11: **else**
12  $\beta_{o\kappa}^{\min} \leftarrow \beta_{o\kappa}$
13: **end if**
14: **end for**
15: **end for**

---

Then the VCOM can be written as,

$$\begin{cases} \max \sum\limits_{\kappa=1}^{m} \sum\limits_{o \in O} F_{ok}(\delta_{o\kappa}, \beta_{o\kappa}) \\ \text{s.t.} \\ (38),\ (41) \sim (43) \\ 0 \leq \gamma_{j\kappa} \leq \sqrt{2}\sigma_{j\kappa}, \kappa \in \kappa_j \\ P_j^{S-} \leq P_{j\kappa}^{S} \leq P_j^{S+}, \kappa \in \kappa_j \\ j = 1, \dots, n \\ \kappa = 1, \dots, m \end{cases} \tag{44}$$

By converting Equation (44) to the unconstrained optimization model as shown in Equation (7), RCGA-rdn can be used to solve the VCOM. The RCGA-rdn stops iterating when the number of the genetic offspring reaches the specified value and we select the best elite solution as the optimal solution.

The ratio plan can be converted into the active power plan of the ESS according to Equation (45):

$$\begin{cases} P_{j\kappa}^{S} = P_j^{S-} + v_j \left( P_j^{S+} - P_j^{S-} \right), \kappa \notin \kappa_j \\ \gamma_{j\kappa} = v_j \sqrt{2}\sigma_{j\kappa}, \kappa \in \kappa_j \end{cases} \tag{45}$$

where $v_j$ is a random number in the range [0, 1].

## 5. Case Study and Discussion

In this section, we verify the effectiveness of the AMM and the proposed day-ahead ESS strategy. The computer used for the test adopts an Intel® Core™ i7-6700 Processor and

8 GB of memory, in which the CPU works at the nominal frequency of 3.4 GHz. The base voltage of the system is 12.66 kV, and the base capacity is 100 MW. The reference voltage is set to 0.95.

Two FCSs are connected to nodes 10 and 25 respectively, as shown in Figure 5. The basic configurations of FCSs A and B are the same as shown in Table 1. The probability distributions of EV load changes of the FCSs A and B at different time intervals in a day are listed in Tables 2 and 3 respectively. Here, sets of charging power test data come from sets of 60-day historical charging power data from two different FCSs. Typical daily EV charging curves of FCSs A and B are shown in Figure 6.

**Table 1.** The basic configuration.

| Parameter | Description | Unit | Value |
|---|---|---|---|
| $S^{Trans}$ | Rated capacity of the service transformer. | kVA | 1000 |
| $S^{PV}$ | Rated capacity of the PV. | kWp | 50 |
| $P^{S+}$ | Rated upper boundary of the charge-discharge active power of the ESS. | kW | 250 |
| $P^{S-}$ | Rated lower boundary of the charge-discharge active power of the ESS. | kW | −250 |
| $SOC_0$ | The initial SOC for the ESS at the beginning of the day. | % | 50 |
| $SOC^+$ | Rated upper boundary of the SOC of the ESS. | % | 30 |
| $SOC^-$ | Rated lower boundary of the SOC of the ESS. | % | 80 |
| $E^S$ | Rated capacity of the ESS. | kWh | 1000 |

**Table 2.** The probability distribution of EV load change at the FCS A in different time intervals in a day.

| $\kappa$ | The EV Load at $\kappa$th Time Interval/kW | | | |
|---|---|---|---|---|
| 1~4 | $N(789,192^2)$ | $N(532,218^2)$ | $N(167,121^2)$ | $E(40)$ |
| 5~8 | $E(40)$ | $E(40)$ | $N(223,40^2)$ | $N(108,74^2)$ |
| 9~12 | $N(107,95^2)$ | $N(190,132^2)$ | $N(294,159^2)$ | $N(406,153^2)$ |
| 13~16 | $N(547,144^2)$ | $N(572,169^2)$ | $N(575,185^2)$ | $N(352,198^2)$ |
| 17~20 | $N(207,163^2)$ | $N(306,145^2)$ | $N(312,163^2)$ | $N(309,180^2)$ |
| 21~24 | $N(328,180^2)$ | $N(350,143^2)$ | $N(494,154^2)$ | $N(690,180^2)$ |

**Table 3.** The probability distribution of EV load change at the FCS B in different time intervals in a day.

| $\kappa$ | The EV Load at $\kappa$th Time Interval/kW | | | |
|---|---|---|---|---|
| 1~4 | $N(789,192^2)$ | $N(532,218^2)$ | $N(167,121^2)$ | $E(40)$ |
| 5~8 | $E(40)$ | $E(40)$ | $E(40)$ | $N(108,74^2)$ |
| 9~12 | $N(107,95^2)$ | $N(190,132^2)$ | $N(294,159^2)$ | $N(406,153^2)$ |
| 13~16 | $N(547,144^2)$ | $N(572,169^2)$ | $N(575,185^2)$ | $N(352,198^2)$ |
| 17~20 | $N(207,163^2)$ | $N(306,145^2)$ | $N(312,163^2)$ | $N(309,180^2)$ |
| 21~24 | $N(328,180^2)$ | $N(350,143^2)$ | $E(40)$ | $N(690,180^2)$ |

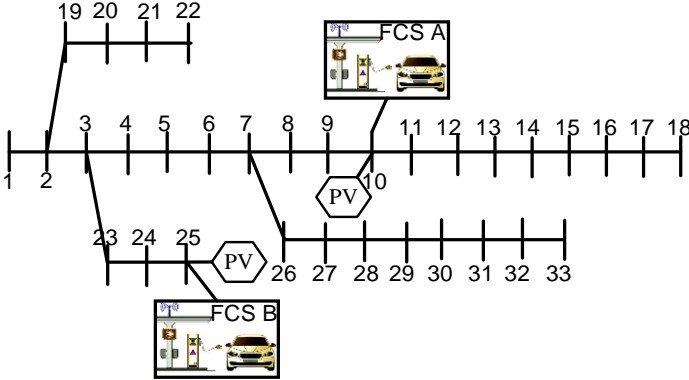

**Figure 5.** IEEE 33-bus test system.

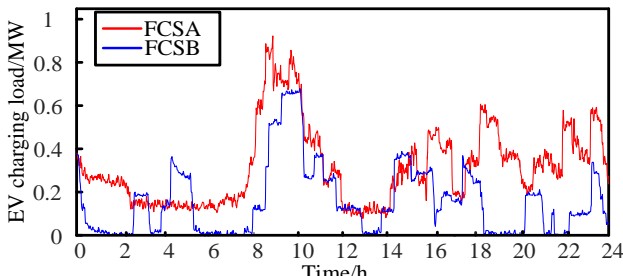

**Figure 6.** Typical daily EV charging curves of FCSs A and B in a day.

Each FCS is equipped with a PV source and the typical daily PV curve is shown in Figure 7.

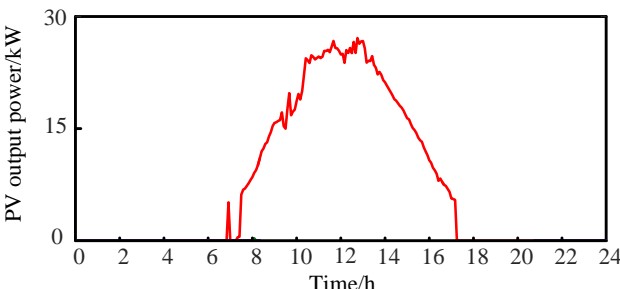

**Figure 7.** Typical daily PV curve in a day.

### 5.1. The Performance of the AMM

The cumulative probability distribution (CPD) of the actual voltage change is obtained by statistical analysis based on a Monte Carlo probability power flow method (MCPFL) [23], which is used as a comparison to the AMM. The sampling times of MCPFL reach 10,000. Figure 8 shows the CPD curves of the voltage change at node 23 at the time interval (17:00, 17:59). It can be seen that the CPD curve obtained by the AMM is very close to the actual CPD curve, which can verify the effectiveness of the AMM. A similar result can also be seen in Figure 9 where node 12 is selected as the observation node. Finally, the correlation coefficient is used to measure the closeness between the actual CPD curve and the CPD curve obtained by the AMM. The corresponding correlation coefficients are listed in Table 4. The correlation coefficients all reach 0.99, which also verifies the effect of the AMM. Table 5 lists the computational overheads of assessment on voltage change at an observation node. It can be seen that the calculation time of the AMM is very short and its CPU occupancy rate is also small.

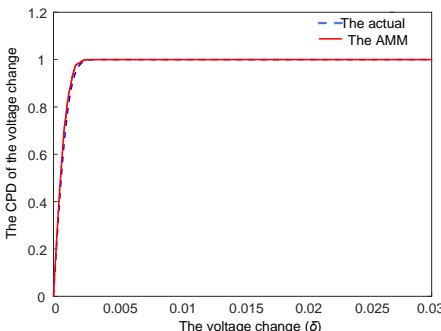

**Figure 8.** The CPD curve of the voltage change of node 23.

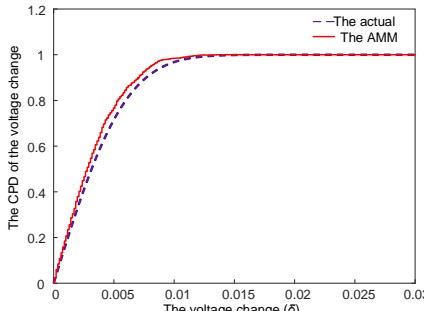

**Figure 9.** The CPD curve of the voltage change of node 12.

**Table 4.** The correlation coefficients.

| Observation Node | The Correlation Coefficient | | | | | | | |
|---|---|---|---|---|---|---|---|---|
| 2~9 | 0.99 | 0.99 | 0.99 | 0.99 | 0.99 | 0.99 | 0.99 | 0.99 |
| 10~17 | 0.99 | 0.99 | 0.99 | 0.99 | 0.99 | 0.99 | 0.99 | 0.99 |
| 18~25 | 0.99 | 0.99 | 0.99 | 0.99 | 0.99 | 0.99 | 0.99 | 0.99 |
| 26~33 | 0.99 | 0.99 | 0.99 | 0.99 | 0.99 | 0.99 | 0.99 | 0.99 |

**Table 5.** Computational overheads.

| Methods | Average Time, s | CPU, % |
|---|---|---|
| MCPFL | 1.527 | 2 |
| AMM | 20.313 | 11 |

*5.2. The Effect of the Proposed Day-Ahead ESS Strategy*

Next, two different cases are studied to verify the proposed day-ahead ESS strategy.

(i)   Case 1: the traditional PV-ESS complementarity strategy is used for the ESS.
(ii)  Case 2: the strategy proposed in this paper is used for the ESS.

For case 1, a compensation control strategy for the ESS mentioned in [12] is used. For case 2, the ESS executes the proposed day-ahead ESS strategy. The ESS automatically enters the charging state both in cases 1 and 2 when the SOC of the ESS is lower than 30%. We select terminal node 18 as an observation node. Besides, the voltage change constraint of the observation node is set to 0.01 and the acceptable probability of the nodal voltage change of the observation node at each time interval is set to 0.95.

Figure 10 shows $P(|\Delta U_o| < \delta_o)$ at each node under different cases in a day. It is obvious that the day-ahead ESS strategy can increase $P(|\Delta U_o| < \delta_o)$ in comparison with the traditional PV-ESS complementarity strategy. Voltages under case 2 are closer to the reference voltage than other cases as shown in Figure 11a, which is more intuitively displayed in Figure 11b. Simulation results above demonstrate the effect of the proposed day-ahead ESS strategy. It can be seen from Figure 12a that the ESS charges during periods of low EV charging load and provides active power compensation by discharge during periods of large EV charging load to reduce the voltage change between actual and reference voltages. In addition, we can find that due to the constraint of the SOC of the ESS, the ESS cannot always compensate for the active power. For example, from 6.00 p.m. to 9.00 p.m., since the SOC of the ESS is close to its lower boundary, the ESS can hardly compensate the active power, which results in the failure of the day-ahead ESS strategy. A similar result can also be obtained according to Figure 13. Therefore, to give full play to the effect of the day-ahead ESS strategy, it is inseparable from reasonable energy storage system planning.

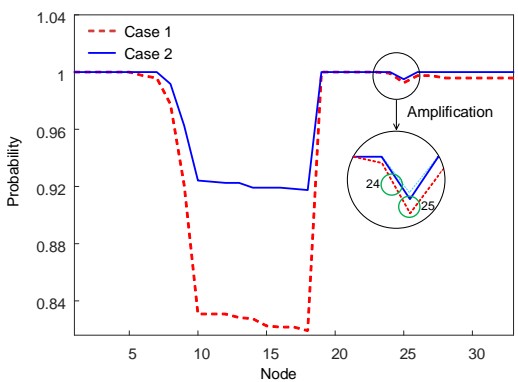

**Figure 10.** P($|\Delta U_o| < \delta_o$) at each node in a day.

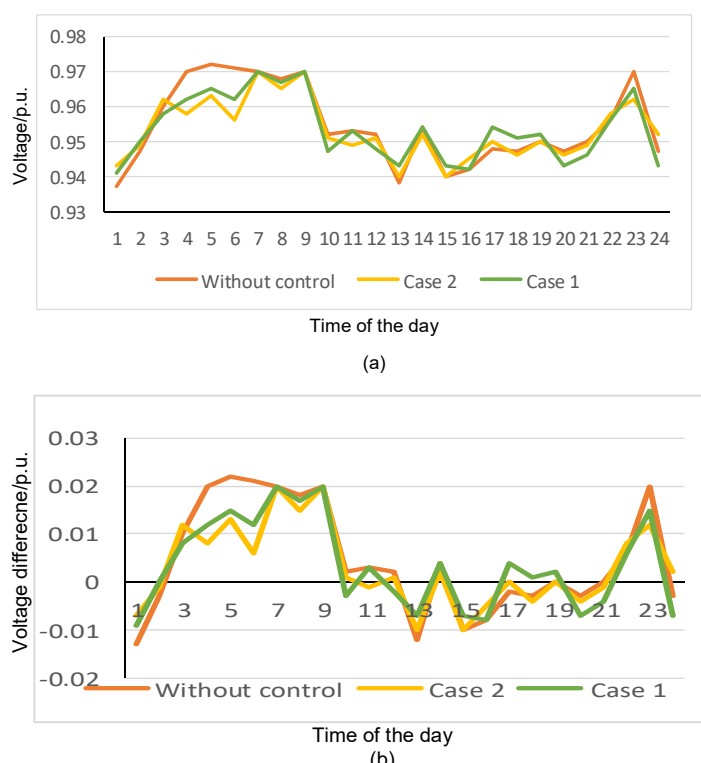

(a)

(b)

**Figure 11.** (**a**) Voltage curves of node 18 under different cases in a day. (**b**) Differences between actual and reference voltages under different cases in a day.

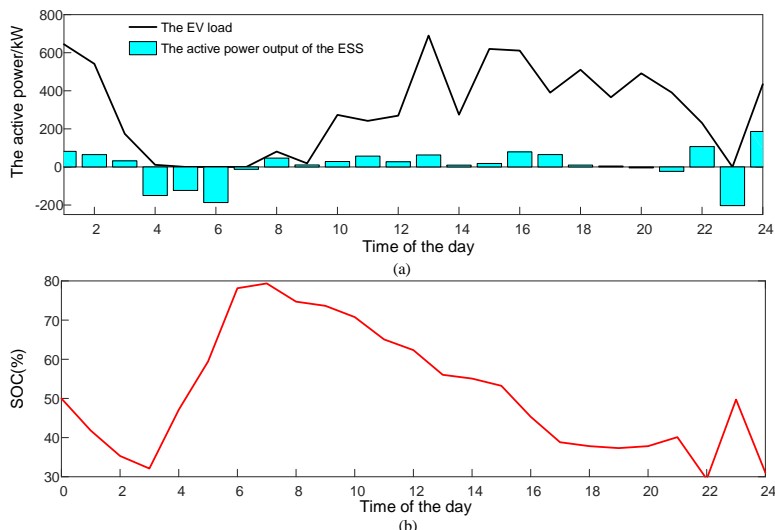

**Figure 12.** (**a**) The active power of the ESS at the FCS A under case 2. (**b**) The SOC of the ESS at the FCS A under case 2.

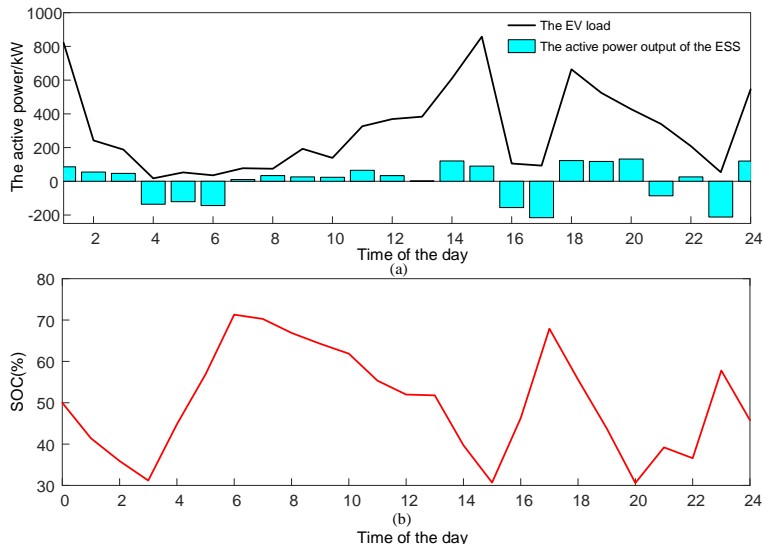

**Figure 13.** (**a**) The active power of the ESS at the FCS B under case 2. (**b**) The SOC of the ESS at the FCS B under case 2.

## 6. Conclusions

With the increasing penetration of EVs, the adverse impact of EV charging load on the nodal voltage in the distribution network can probably be aggravated. To alleviate this impact, this paper proposes a centralized day-ahead ESS strategy with full consideration of the randomness of EV charging load. This strategy can be deployed in the RODS and its output can be calculated based on the VCOM. Finally, simulation results verify the effectiveness of the proposed day-ahead ESS strategy. The main contributions of this article are as follows:

- A voltage change optimization model (VCOM) considering the randomness of the EV load is constructed to alleviate the voltage change problem caused by EV fast charging and it can be easily solved by traditional intelligent algorithms, such as GA.
- An analytical assessment model (AAM) of the nodal voltage change with shorter computational time and higher reliability is proposed.

**Author Contributions:** Conceptualization, R.Y., X.H. and Z.Y.; data curation, R.Y. and Z.Y.; formal analysis, R.Y.; methodology, R.Y.; resources, R.Y.; software, R.Y.; validation, R.Y. and Z.Y.; writing—original draft, R.Y. All authors have read and agreed to the published version of the manuscript.

**Funding:** This research received no external funding.

**Conflicts of Interest:** No conflict of interest exists in the submission of this manuscript, and the manuscript is approved by all authors for publication. I would like to declare on behalf of my co-authors that the work described was original research that has not been published previously, and is not under consideration for publication elsewhere, in whole or in part. All the authors have approved the manuscript that is enclosed.

## Nomenclature

| | |
|---|---|
| $P_{j\kappa}^{\mathrm{G}}$ | Total grid load at the FCS j at $\kappa$th time interval. |
| $\mu_{j\kappa}^{\mathrm{G}}$ | Expectation of $P_{j\kappa}^{\mathrm{G}}$. |
| $\gamma_{j\kappa}$. | Standard change of $P_{j\kappa}^{\mathrm{G}}$ |
| $P_{j\kappa}^{\mathrm{PV}}$ | Output power of the PV at the FCS $j$ at $\kappa$th time interval. |
| $P_{j\kappa}^{\mathrm{S}}$ | Charge–discharge active power of the ESS at the FCS $j$ at $\kappa$th time interval. |
| $P_{j\kappa}^{\mathrm{EV}}$ | The NFCL at the node $j$ with the FCS at $\kappa$th time interval. |
| $P_{j\kappa}^{\mathrm{EV}}$ | The EV's charging load at the FCS $j$ at $\kappa$th time interval. |
| $\mu_{j\kappa}$ | Expectation of $P_{j\kappa}^{\mathrm{EV}}$. |
| $\sigma_{j\kappa}$ | Standard deviation of $P_{j\kappa}^{\mathrm{EV}}$. |
| $S_j^{\mathrm{Trans}}$ | Rated capacity of the service transformer at the FCS $j$. |
| $\delta_{o\kappa}$ | Nodal voltage change limit at the observation node o at $\kappa$th time interval. |
| $\alpha_{o\kappa}$ | Shape parameters for the nodal voltage change at observation node $o$ at $\kappa$th time interval. |
| $\beta_{o\kappa}$ | Control parameter for the nodal voltage change at observation node $o$ at $\kappa$th time interval. |
| $x_\kappa$ | Optimization variable at $\kappa$th time interval. |
| $pp$ | Initial population number. |
| $U_s$ | Phase voltage at the source node $s$. |
| $U_d$ | Phase voltage at the load node $d$. |
| $U_o$ | Phase voltage at the observation node $o$. |
| $S_d^*$ | Conjugate complex draw power at node $d$. |
| $U_d^*$ | Conjugate phase voltage at node $d$. |
| $Z_{od}$ | Shared impedance between node $d$ and observation node $o$ from the source node $s$. |
| $P_j^{\mathrm{S}+}$ | Rated upper limit of the charge-discharge active power of the ESS at the FCS $j$. |
| $P_j^{\mathrm{S}-}$ | Rated lower limit of the charge-discharge active power of the ESS at the FCS $j$. |
| $P_{j\kappa}^{\mathrm{Sd}}$ | Dynamic lower limit of the charge-discharge active power of the ESS at the FCS $j$ at the $\kappa$th time interval. |
| $P_{j\kappa}^{\mathrm{S}}$. | Active power output of the ESS at the FCS $j$ at $\kappa$th time interval. |
| $\xi_j^{\mathrm{Ps}}$ | A preset constant for restraining $P_{j\kappa}^{\mathrm{S}}$. |
| SOC | State of charge. |
| $\mathrm{SOC}_{j0}$ | The initial SOC of the ESS at the FCS $j$ at the beginning of the day. |
| $\xi_j^{\mathrm{SOC}}$ | A preset constant for restraining the SOC for the FCS $j$. |
| $SOC_j^+$ | Rated upper limit of the SOC. |
| $SOC_j^-$ | Rated lower limit of the SOC. |
| $SOC_{j\kappa}$ | Value of the SOC of the ESS at the FCS $j$ at $\kappa$th time interval. |
| $\Gamma(.)$ | Gamma function. |
| $\Phi(.)$ | Normal probability distribution function. |

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
