# Peer review of "Energy Management Method for Fast-Charging Stations with the Energy Storage System to Alleviate the Voltage Problem of the Observation Node"

_wevj, doi:10.3390/wevj12040234_

Round 1

Reviewer 1 Report

The authors proposed a day-ahead scheduling strategy to mitigate voltage security challenges in power distribution nodes resulting from connected energy storage systems and fast-charging stations. The authors gave a good introduction with sufficient background study to the problem. They also presented adequate theoretical and technical formulations of the strategy. The authors verified the effectiveness of the proposed strategy by applying it to the IEEE 33 Bus system and presented their results which suggest that the proposed strategy is effective. Such a topic as is in this manuscript is vital and timely considering the wide acceptance and fast adoption of electric vehicles and the consequent springing up of fast-charging stations to service these vehicles.

However, I recommend that the authors should revisit the manuscript and do a thorough English Language review and editing of the report, especially the Abstract and the Introduction sections, which were grammatically poorly written. Also, ensure that every acronym used is accompanied by its full meaning when it is first used. (See line 11.) These recommended reviews and editing will make the manuscript more readable.

Author Response

Thank you for your comments. We have solved these mistakes. Please check the modifications in the revised paper.

       In all, we hope that the above modifications and interpretations can satisfy the reviewer. Special thanks for your good comments.

Reviewer 2 Report

The paper makes an important contribution to a research area of growing importance.

The figures are very clear

The referencing is appropriate and the article is well referenced

I found no issues with the mathematical modelling calculations

The only question that I have relates to the statement in lines 134-135 in which it is stated that 

The EV’s charging load changes at the 4th, 5th and 7th time intervals and are concentrate within 20 kW. From Figure 4, the EV charging loads appear to be around 40 kW

The main problem with the paper is the English. I have made several suggested changes to the wording below, but I may have missed some of the wording or grammatical errors

Line 8 to 9.

“The large-scale fast charging of electric vehicles (EVs) may cause the nodal voltage to change large in the distribution grid, which can give the adverse impact on the normal operation of the power load.”.

Should be

“Large-scale fast charging of electric vehicles (EVs) may cause large changes in nodal voltages in distribution grids, which can have adverse impacts on the normal operation of power loads.”

Line 11.

The meaning of the acronym FCS (fast charging station) needs to be added here as it is the first time in the paper that the acronym is used.

Lines 11-13

“ … which is aimed at maximizing the probability of the nodal voltage change smaller than a pre-set limit at the observation node so that reducing the nodal voltage change caused by the FCSs.”

Should be

“ … with the aim of maximizing the probability of the nodal voltage change being smaller than a pre-set limit at the observation node, thereby reducing the nodal voltage change caused by the FCSs.”

Line 15.

“Besides, a voltage change optimization … “

Should be

“Furthermore, a voltage change optimization … “

Lines 25-27

“With the growing environmental problem, significant attention is given on developing EVs which is regarded as a tremendous potential measure to contribute to reducing the carbon emissions [1], [2].”

Should be

“The adoption of EVs is regarded as having very large potential in terms of reducing carbon emissions [1], [2].”

Lines 27-28.

“In recent years, the number of EVs grows rapidly, e.g., the worldwide possession quantity of EVs has surpassed 5 million by 2018 [3]”

Should be

“The take up rate of EVs has increased rapidly in recent years, the number of EVs registered worldwide by 2018 having surpassed 5 million.”

Lines 30-32.

”During peak load period, the distribution grid may easily suffer security risk, such as transformer overload, voltage problems, cable overload, etc. [6].”

Should be

”During peak load periods, distribution grids may suffer security risks, such as transformer overload, voltage problems or cable overload [6].”  [Note: t is not correct to use both ‘such as” and “etc.” in a single sentence. Just one or the other should be used]

Line 33

“The voltage problems caused by the increase of EV’s charging are widely concerned.”

Should be

“The voltage problems caused by increasing EV charging loads are widely recognised.”

Lines 34-35

“[7] confirms that high-level penetration of EVs can reduce the voltage stability margin and may lead to the voltage instability of the power gird.”

Should be

“Previous studies have confirmed that high penetrations of EVs can reduce the voltage stability margin and may lead to the voltage instability of the power gird [7].”

Lines 35-36

“In [8], a similar conclusion is obtained by using the IEEE 3-bus test system”

Should be

“A similar conclusion was obtained by using the IEEE 3-bus test system [8]”.

Lines 36-38

“Since in practical the voltage instability barely takes place in the low-voltage (LV) distribution network, more researchers focus on the impact 37 of EV’s charging on voltage change.”

Should be

“Since there is very little voltage instability in the low-voltage (LV) distribution networks operating under normal conditions, more research focus is required on the impacts of EV charging on voltage change.”

Lines 38-40

“The randomness and impulsive characteristics of EV loads make the injection power at the charging node vary greatly, which may enlarge nodal voltage change at the local node or neighboring nodes.”

Should be

The randomness and impulsive characteristics of EV recharging loads means that the injection of power at the charging node van vary greatly, which may increase the nodal voltage change at the local node or neighboring nodes.”

Lines 45 to 47

“Among [7]-[12], the traditional Newton power flow method and the Monte Carlo method are the main resorts to analysis the voltage problems.”

Should be

“The traditional Newton power flow method and the Monte Carlo method are the main types of  analysis of the voltage problems [7]-[12].”

Lines 57-58

“Therefore, the indicator with cheap calculation and enough precision is still need to be further study.”

Should be

“Therefore, a method of analysing the voltage problem that has low calculation requirements while having sufficient precision is still needed.”

Lines 69-71

“To address the inadequacy, local grid load control methods are proposed, which attempt to impose restrictions on nodal load to ensure the nodal voltage within endurable range”

Should be

“To address this inadequacy, local grid load control methods have been proposed that attempt to impose restrictions on the nodal load to ensure that the nodal voltage remains within an acceptable range”

Lines 83-84

“Unfortunately, the related researches can 83 hardly be found [23].”

Should be

“Unfortunately, there has been little research effort undertaken in this area [23].”

Line 84

“In attempt to fill the inadequacy … ”

Should be

“In an attempt to fill this inadequacy … ”

Line 116

“, then the FCS model can be expressed as,”

Should be

“, in which case the FCS model can be expressed as,”

Line 131

“We equally divide a day into 24 time intervals, ”

Should be

“We divide a day into 24 equal time intervals, ”

Lines 134-135

“The EV’s charging load changes at 4th, 5th and 7th time intervals concentrate within 20kW, which can hardly give the nodal voltage significant impact under normal circumstances.”

Should be

“The EV’s charging load changes at the 4th, 5th and 7th time intervals and are concentrate within 20 kW, which would not have a significant impact on the nodal voltage under normal circumstances.”

Lines 157-160.

There appears to be a change in font size in these lines relative to the previous text

Lines 202

“The bigger P(|∆Uo|<δo) is, the smaller the impact of the FCSs on the 202 voltage change will be. > is,”

Should be

“The larger P(|∆Uo|<δo) is, the smaller the impact of the FCSs on the 202 voltage change will be. > is,”

Line 203

“we derive The AEF-vc which is used to evaluate”

Should be

“we derive the AEF-vc, which is used to evaluate”

Line 261

“In this section, we verify the effectiveness of The AEF-vc … ”

Should be

“In this section, we verify the effectiveness of the AEF-vc … ”

[Note: This same error occurs in several other places in the document]

Line 289

“Besides, when the voltage change limit (δ) is”

Should be

“Furthermore, when the voltage change limit (δ) is”

Line 374

“Besides, the voltage sensitivity indicators at nodes 24 and 25 under case”

Should be

“Furthermore, the voltage sensitivity indicators at nodes 24 and 25 under case”

Author Response

Thanks for your comments. According to your good suggestion, we have corrected English grammar mistakes. In addition, we have deleted Fig. 4 since this figure may mislead readers. Please check the modifications in the revised paper.

       We sincerely hope that the modifications can be acknowledged by the reviewer. Special thanks for your good comments.

Reviewer 3 Report

In my opinion, the manuscript entitled "Energy Management Method for the Fast Charging Station with the Energy Storage System to Alleviate the Local Under-voltage Problem" can be published in WEVJ after major revision.

The article is interesting, but there are mistakes.

Details of my revision are in the attached file.

Author Response

Thanks for your recommendation for publication and your comments. Mistakes you mentioned have been corrected.

       Special thanks for your good comments. We sincerely hope that the modifications can be acknowledged by the reviewer.

Reviewer 4 Report

The paper provides new energy management method for the fast charging station with the energy storage system. This is very relevant topic due to fast penetration of renewables requiring energy storage especially in transport sector where number of electric vehicles  is constantly growing. Proposed case study is very useful however paper needs improvements in the structure.  The section on methods and data is missing. This is necessary for scientific papers. Another issue is pour references. There are man new studies in this field and authors should build they study on their analysis by providing gap necessary to fill. Discussion section needs to be separated from case study analysis. The results should be discussed in the light of other studies conducted and compared. The main advantages of proposed  approach need to be highlighted. limits also  need to be addressed. The main findings need to be highlighted in conclusions. Conclusions  need to b e better structured.

Author Response

Thanks for your comments. Firstly, we have added some new references about the impact of fast charging stations on nodal voltages in the distribution network as follows:

[12]J., Deng; J., Shi; Y., Liu. Application of a hybrid energy storage system in the fast charging station of electric vehicles. IET Generation, Transmission & Distribution 2016, 10(4), 1092-1097.

[13] Muntaser, A. ; Elwarfalli, H. ; Kumar, J. ; Subramanyam, G. Development of advanced energy storage system using fuzzy control. Aerospace & Electronics Conference. IEEE 2017.

[14] Kornsiriluk, Varawut. Study of Energy Storage System: Concept of Using ESS in EV Charging Stations in MEA. 2019 IEEE PES GTD Grand International Conference and Exposition Asia (GTD Asia) IEEE 2019.

Secondly, the structure of this paper have been reorganized. We have moved Section 3.4 to Section 5 so that the discussion section can be separated from the case study analysis. Thirdly, inadequacies in current studies have been highlighted in the introduction. Finally, we have rewritten Section 5 and main contributions of this paper have been listed in the conclusion. Please check the modifications in the revised paper.

       Special thanks for your good comments. We sincerely hope that the modifications can be acknowledged by the reviewer.

Reviewer 5 Report

The paper presents day-ahead scheduling of energy storage system to maintain the nodal voltage within stable limits which would otherwise have been affected by fast charging stations. The topic is interesting. However, the paper lack novelty and it has been poorly written. Even the abstract has many grammatical errors. The paper should be proofread by professional English writer. In terms of originality, impact of fast charging stations on distribution system voltages has already been studied. Please improve the literature survey and present strong contributions.  

Author Response

Thanks for your comments. Firstly, we have added some new references about the impact of fast charging stations on nodal voltages in the distribution network as follows:

[12]J., Deng; J., Shi; Y., Liu. Application of a hybrid energy storage system in the fast charging station of electric vehicles. IET Generation, Transmission & Distribution 2016, 10(4), 1092-1097.

[13] Muntaser, A. ; Elwarfalli, H. ; Kumar, J. ; Subramanyam, G. Development of advanced energy storage system using fuzzy control. Aerospace & Electronics Conference. IEEE 2017.

[14] Kornsiriluk, Varawut. Study of Energy Storage System: Concept of Using ESS in EV Charging Stations in MEA. 2019 IEEE PES GTD Grand International Conference and Exposition Asia (GTD Asia) IEEE 2019.

Besides, we have listed main contributions of this paper in the conclusion. Please check the modifications in the revised paper.

       Special thanks for your good comments. We sincerely hope that the modifications can be acknowledged by the reviewer.

Round 2

Reviewer 3 Report

Accept in present form

Author Response

Thanks for your comments

Reviewer 4 Report

The authors have revised their paper and integrated my all comments. The answers to my comments are provided. The current improved version of manuscript can  be published.

Author Response

Thanks for your comments

Reviewer 5 Report

Most of the comments are still not addressed.

Literature survey in the introduction is should be comprehensive but concise. However, Literature review has still not been remarkably improved. Please include most recent published articles on EV fast charging charging stations and their impact in power system.

The contributions are still weak. The impact analysis of EVs FCS on distribution system is already available in the literature. What is the new contribution in this paper.

Management of range anxiety of EVs is neccessary for effective participation of EV users. However, it has not been considered.

In Figure 7, The output power of PV system is high. What would happen in case of cloudy. How EVs FCS will contribute to the distribution system?

In Figure 11, voltage is well below 0.95 pu for considerable time of the day. EVs FCS are not helping in improving distribution system voltage. How EVs FCS should be placed in distribution system that this problem can be rectified.

Figure  12 shows that SOC of ESS is at minimum level at the end of the day. Howe will it impact next day charging of EVs. How range anxiety of EVs has been justified here?

Author Response

Dear Reviewer 5#:

Thank you for your comments.

  1. Response to comment: (iterature survey in the introduction is should be comprehensive but concise. However, Literature review has still not been remarkably improved. Please include most recent published articles on EV fast charging charging stations and their impact in power system.)

Response: This paper focuses on the adverse impact of EV charging on the nodal voltage in the distribution network and proposes an ESS strategy to alleviate this impact. Therefore, we only listed references related to this topic. In our opinion, the range anxiety problem and the nodal voltage problem are two separate issues, so we have not listed references related to the range anxiety problem.

  1. Response to comment: (The contributions are still weak. The impact analysis of EVs FCS on distribution system is already available in the literature. What is the new contribution in this paper.)

Response: Many studies show that installing the energy storage system (ESS) in the fast charg-ing station with the appropriate energy management strategy can alleviate voltage prob-lems. A compensation control strategy for the ESS is proposed to deal with energy imbal-ance in continuous operation [12]. An advanced ESS control strategy based on fuzzy con-trol is used to reduce the local voltage fluctuation [13]. [14] studies the ESS used in the FCS, which demonstrates that the voltage profile of the transformer in the FCS can be improved by the ESS. A local control algorithm for the FCS with flywheel energy storage systems is proposed in [15], which can also guarantee the charging demand power of the EV during the period of charging control. However, the randomness of the EV load is not considered in these studies. Furthermore, the actual adjustable capacity of the ESS in a FCS is very limited so that it is generally less effect through using above local voltage regulation methods to solve the voltage problem in the distribution network.

To address these inadequacies, a centralized day-ahead scheduling strategy for ESSs in FCSs is proposed in this paper to reduce the adverse impact of FCSs on the voltage changes at observation nodes in the radial distribution network. The main contributions of this paper are as follows:

  • A voltage change optimization model (VCOM) considering the randomness of the EV load is constructed to alleviate the voltage change problem caused by EV fast charging and it can be easily solved by traditional intelligent algorithm, such as GA.
  • An analytical assessment model (AAM) of the nodal voltage change with shorter computational time and higher reliability is proposed.

  1. Response to comment: (Management of range anxiety of EVs is neccessary for effective participation of EV users. However, it has not been considered.)

Response: In this paper, the proposed ESS strategy is used to regulate the charging and discharging active power of the ESS installed in the FCS so that voltage differences of concerned nodes can be reduced. Therefore, the charging demand power of EV is not limited under this ESS strategy.

  1. Response to comment: (In Figure 7, The output power of PV system is high. What would happen in case of cloudy. How EVs FCS will contribute to the distribution system?)

Response: Since we proposed an ESS strategy, the cloudy weather will not affect the execution of the strategy.

  1. Response to comment: (In Figure 11, voltage is well below 0.95 pu for considerable time of the day. EVs FCS are not helping in improving distribution system voltage. How EVs FCS should be placed in distribution system that this problem can be rectified.)

Response: Since there are only two FCSs in the simulation, to reflect the effect of the strategy, the distribution network is set in a heavy load state. The objective of this ESS strategy is to reduce the difference between the reference voltage (0.95p.u.) and current voltage. Voltages under case 2 are closer to the reference voltage than other cases as shown in Fig. 11 (a), which is more intuitively displayed in Fig.11 (b).

  1. Response to comment: (Figure 12 shows that SOC of ESS is at minimum level at the end of the day. Howe will it impact next day charging of EVs. How range anxiety of EVs has been justified here?)

Response: In real world, ESS can be actively charged to a specified SOC value by a staff member. Therefore, it is reasonable to ignore the impact of the SOC value on the strategy execution on the next day.

Special thanks for your comments. We sincerely hope that responses can be acknowledged.